# Characterization and Potential Food Applications of Oat Flour and Husks from Differently Colored Genotypes as Novel Nutritional Sources of Bioactive Compounds

**DOI:** 10.3390/foods13233853

**Published:** 2024-11-28

**Authors:** Valentina Nikolić, Slađana Žilić, Marijana Simić, Katarina Šavikin, Tatjana Stević, Jelena Živković, Beka Sarić, Danka Milovanović, Vesna Kandić Raftery

**Affiliations:** 1Research Department, Maize Research Institute, Zemun Polje, Slobodana Bajića 1, 11080 Belgrade, Serbia; szilic@mrizp.rs (S.Ž.); marijana.simic@mrizp.rs (M.S.); bsaric@mrizp.rs (B.S.); dmilovanovic@mrizp.rs (D.M.); 2Institute of Medicinal Plants Research, Dr. Josif Pančić, Tadeuša Koščuška 1, 11000 Belgrade, Serbia; ksavikin@mocbilja.rs (K.Š.); tstevic@mocbilja.rs (T.S.); jzivkovic@mocbilja.rs (J.Ž.); 3Breeding Department, Maize Research Institute, Zemun Polje, Slobodana Bajića 1, 11080 Belgrade, Serbia; vkandic@mrizp.rs

**Keywords:** oats, chemical composition, phenolic compounds, antioxidant capacity, antimicrobial activity, human gut microbiota

## Abstract

Oats are gluten-free cereals rich in dietary fiber, *β*-glucans, phenolic acids, flavonoids, carotenoids, vitamin E, and phytosterols. They have been used in traditional medicine for centuries to treat hyperacidity, acute pancreatitis, burns, and skin inflammation. This study assessed the nutritional and phenolic profile of oat flour (OF) and ground oat husks (OHs) from white, brown, and black hulled oat genotypes, as well as the antioxidant and antimicrobial activity of their extracts. The extracts were tested on six strains of gastrointestinal tract pathogens. OF samples had, on average, a high protein content (15.83%), fat content (6.27%), and *β*-glucan content (4.69%), while OH samples were rich in dietary fiber. OHs had significantly higher average total phenolic compounds compared to OF and had twice as high antioxidant capacity. Ferulic acid was predominant in all samples, followed by *p*-coumaric, isoferulic, vanillic, and syringic acid. The traditionally prepared OH extracts manifested the best bactericidal activity against *Listeria monocytogenes*, *Escherichia coli*, and *Staphylococcus haemolyticus*, while *Salmonella typhimurium* was the least sensitive to the bactericidal effect of all the investigated samples. Both OF and powdered OHs have potential applications in the functional food industry and pharmacy due to their bioactive compounds, their biological activity, as well as their overall nutritional profile.

## 1. Introduction

Hippocrates’ maxim, “Let food be thy medicine and medicine be thy food,” which dates back two millennia, was nearly forgotten in the 19th century due to the development of modern medicine. However, a renewed focus on the role of nutrition in promoting health and preventing disease by utilizing plants with medicinal properties has regained popularity in the past several decades [1]. Epidemiological studies have shown that the consumption of whole-grain cereals is associated with a decreased risk of some chronic diseases and conditions such as diabetes [2] cardiovascular disease [3], and obesity [4]. Their beneficial activity is mostly connected to the presence of dietary fiber, vitamins, essential fatty acids, and phenolic compounds [5].

Oat (*Avena sativa* L.) is the seventh most extensively grown and commercially significant cereal in the world. It was first cultivated for its medicinal properties, before being employed as a nutrient-dense food source for humans and animals [6]. Oats exhibit a broad range of biological activity, which suggests their potential therapeutic value [7,8]. Namely, oats have been found to promote physiological benefits such as reducing hyperglycemia, hyperinsulinemia, hypercholesterolemia, hypertension, and cancer [9]. Oats have been used in European, Chinese, and Middle Eastern traditional medicine for centuries to treat mental and physical ailments [6,10,11], as well as obesity, constipation, loss of appetite, and headaches [12,13,14]. In Serbian traditional medicine, oat grains are used as a remedy for hyperacidity and acute pancreatitis, and as a treatment for burns and skin inflammation. Furthermore, oat straw is frequently used to treat frostbite, sciatica, and rheumatism [15].

Oats are naturally gluten-free grains suitable for persons with gluten-related disorders, including celiac disease [16]. They are predominantly composed of starch, which makes up more than 60% of the grain weight and is crucial in determining oats’ physicochemical and structural characteristics [17]. Whole-grain oats are rich in beneficial macronutrients, such as lipids with a high degree of unsaturated fatty acids, which make up about 40 and 36% of total fatty acids, respectively, and proteins with a well-balanced essential amino acid composition [18]. Oats contain dietary fiber with a high level of *β*-glucan (2–8.5% *w/w* of oat grain), a soluble form of fiber that has been shown to have a variety of physiological effects, including controlling postprandial blood glucose levels and lowering serum cholesterol levels [19,20]. Nevertheless, oat grains are abundant in bioactive compounds such as avenanthramides, polyphenols, phenolic acids, carotenoids, vitamin E, and phytosterols [6,8,21].

The inedible outer husks of the oat grain, typically removed during harvest and processing, are the main byproduct of oat production and can make up as much as 35% of the grain [22]. Oat husks (OHs) are frequently underestimated as agro-industrial wastes and are commonly fed directly to animals or disposed of in landfills [23]. OHs also have a range of biologically important components, primarily proteins (1–7%, dry weight basis), cellulose (16–26%), hemicellulose (24–35%), and lignin (13–25%) [22,23]. OHs are a rich source of bioactive compounds, including phenolic acids and avenanthramides, which have many health-promoting qualities. The most prevalent phenolic components in oats are phenolic acids, mostly present in the three-layer bran (peel, seed coat, and aleurone) [24]. The consumption of phenolic acids has been associated with lowering the risk of cardiovascular disease [25]. In addition to their potential health benefits, phenolic acids have been demonstrated to improve gut health by promoting the growth of beneficial bacteria and suppressing pathogens, which supports overall digestive health [26,27]. Only a small portion of the phenolic compounds are immediately absorbed by the small intestine; the gut microbiota will metabolize up to 90% of these compounds in the colon [28]. Accordingly, the breakdown and metabolism of phenolic compounds are greatly influenced by the gut microbiota [29]. However, aside from using them as a source of dietary fiber, the OH utilization level is still considerably low [23].

The prebiotic impact of dietary phenolic compounds on gut microbiota has been demonstrated in numerous in vitro and in vivo investigations in recent years [30,31]. For instance, adding bound phenolic compounds from rice bran to human fecal homogenate in vitro typically suppresses the growth of harmful bacteria [32]. According to Gong et al. [30], higher intakes of phenolic acids from whole wheat were linked to increased abundances of *Bifidobacterium* and *Lactobacillus* and lower abundances of *Escherichia coli, Clostridiaceae*, and *Clostridium perfringens*. Furthermore, oat *β*-glucan is a prebiotic that modulates intestinal flora and aids in the treatment of diarrhea and related conditions by preserving the energy balance thanks to several significant physicochemical properties, including solubility, viscosity, and gelation [19]. Human endogenous digestive enzymes and the gastric acid environment scarcely hydrolyze oat *β*-glucan; instead, it enters the large intestine and is broken down by gut bacteria [19].

The objective of this study was to assess the nutritional and phenolic profile of oat flour and ground oat husk obtained from black, brown, and white hulled oat genotypes, as well as the antioxidant and antimicrobial activity of their extracts rich in bioactive compounds in order to evaluate their potential for the production of functional foods and pharmacological application.

## 2. Materials and Methods

### 2.1. Chemicals and Consumables

The HPLC grade chemicals–6-Hydroxy-2,5,7,8-tetramethylchroman-2-carboxylic acid (Trolox), 2,2′-azino-bis/3-ethylbenzothiazoline-6-sulfonic acid (ABTS), and phenolic acid standards: gallic acid, 3,4-dihydroxybenzoic acid, chlorogenic acid, vanillic acid, caffeic acid, syringic acid, *p*-coumaric acid, sinapic acid, ferulic acid, and isoferulic acid–were purchased from Sigma-Aldrich (Steinheim, Germany). Formic acid (98%) and methanol were purchased from J.T. Baker (Deventer, The Netherlands). The p.a. grade Folin-Ciocalteu reagent was procured from Sigma-Aldrich (Steinheim, Germany). The p.a. grade chemical potassium persulfate (dipotassium peroxdisulfate) was bought from Fluka Chemie AG (Buchs, Switzerland). Sodium hydroxide, sulfuric acid, ethyl acetate, diethyl ether, and ethanol were purchased from Merck (Darmstad, Germany). Sodium tetraborate-10-hydrate and boric acid were bought from Sigma-Aldrich (St. Louis, MO, USA). Assay kits for the determination of resistant starch (K-RSTAR) and *β*-glucan (K-BGLU) contents were purchased from Megazyme (Wicklow, Ireland). The enzymes used for the in vitro digestion protocol were all procured from Sigma-Aldrich, Merck, namely: pepsin from porcine gastric mucosa (P7000-25G), bile extract porcine (B8631-100G), pancreatin from porcine pancreas (P1750-25G), protease from *Streptomyces griseus* (P5147-1G), and viscozyme L cellulolytic enzyme mixture (V2010-50ML). Syringe filters (nylon, 0.45 mm) and Hypersil GOLD aQ C18 column (150 mm × 4.6 mm, i.d., 3 µm) were supplied by Thermo Fisher Scientific (Waltham, MA, USA). Ultrapure water was used throughout the experiments (LaboStar Pro system, Evoqua, Pittsburgh, PA, USA).

### 2.2. Plant Material

The experimental material comprised three oat (*Avena sativa* L.) genotypes obtained from the Maize Research Institute, Zemun Polje (MRIZP) gene bank. The genotypes were chosen based on differences in agronomic traits such as yield and its components, as well as the grain husk color. Single genotypes of yellow, brown, and black oats were cultivated at the location of MRIZP Zemun Polje, Serbia (44°52′ N, 20°19′ E, 82 m a.s.l.), sown in the growing season of 2023/2024. Standard cropping practices were applied to provide adequate nutrition and to keep the plots disease- and weed-free. After harvesting, the broken and damaged grains, as well as extraneous matter, were removed from the samples. The oat grains were hulled using a scalpel to separate the OHs and groats manually. The separated OHs and groats were ground on a Perten 120 lab mill (Perten Instruments AB, Hägersten, Sweden) to obtain fine powder (particle size < 500 μm) OF and OHs for the analyses. The oat grain is referred to as OF (oat flour) throughout the manuscript for simplicity reasons. All the prepared samples were stored at −70 °C before analysis.

### 2.3. Chemical Procedures

#### 2.3.1. Analysis of Basic Chemical Composition

The dry matter content of the samples was assessed by using the conventional drying method in an oven at 105 °C to a constant mass. The total starch content was determined on UniPol L 2020 polarimeter (Schmidt + Haensch GmbH and Co., Berlin, Germany) according to the Ewers polarimetric method [33]. The total protein content was determined by the standard micro- Kjeldahl method (AOAC 920.87) (AutoKjeldahl distillation unit K-350 and speed digester K-439, BÜCHI Labortechnik, Flawil, Switzerland) as the total nitrogen multiplied by 5.7 [34]. The fat content was determined according to the standard Soxhlet method (AOAC 920.39) [35] on a FatExtractor E-500 (Büchi Labortechnik, Flawil, Switzerland). The ash content was analyzed by the slow combustion of the sample at 550 °C in a muffle furnace (L47, 1200 °C, Naber Industrieofenbau, Lilienthal, Germany) by following the AOAC 923.03 method [35]. All the results are given as means ± standard deviation of three repetitions and expressed as a percentage per dry matter (d.m.).

#### 2.3.2. Analysis of the Alcohol-Soluble Protein Fraction

The ethanol-soluble protein fraction was obtained by successive extractions of OF and ground OHs with a series of solvents (in a ratio of 1:10 *w/v*) according to the Osborne procedure, as described by Lookhart and Bean [36], with some modifications. Distilled water, 0.5 M NaCl, and 70% ethanol were used to extract albumin, globulin, and prolamin fractions. Except for the albumin fraction, all the others were extracted from the water-washed pellet remaining after the previous extraction. The extraction of each protein fraction was performed by repeated stirring three times for 30 min at room temperature, followed by centrifugation at 10,000 rpm for 5 min at 4 °C. The final volume of each protein extract was 50 mL. The protein extracts (10 mL) were evaporated for 12 h at 100 °C. The protein content in the ethanol-soluble fraction was analyzed after the micro Kjeldahl method [35] on the BÜCHI Kjeldahl System (Auto Kjeldahl Distillation Unit K-350 and Speed Digester K-439, BÜCHI Labortechnik, Flawil, Switzerland) and calculated from the nitrogen content determined, using 5.7 as the conversion factor. The results are given as the percentage of the d.m. and the percentage of the total protein (protein solubility index).

#### 2.3.3. Analysis of Dietary Fibers

The contents of hemicellulose, cellulose, neutral detergent fibers (NDF), acid detergent fibers (ADF), and lignin (ADL) were determined by the Van Soest detergent method using the Fibertec system 2010 (Foss, Hillerød, Denmark). The procedure is described in detail in a previous paper [37]. The method is based on the fibers’ solubility in neutral, acid, and alkali reagents. NDF practically represents total insoluble fibers (not soluble in water); ADF mainly consists of cellulose and lignin; and ADL is pure lignin. The hemicellulose content was obtained as a difference between the NDF and ADF contents, while the cellulose content was calculated as the difference between the ADF and lignin contents. All the results are given as the percent per d.m.

#### 2.3.4. Analysis of β-Glucan

The content of *β*-glucan was determined using the Megazyme protocol [38]. Briefly, samples (0.5 g) were suspended and hydrated in ethanol (1 mL, 50% *v/v*) and sodium phosphate buffer solution (4 mL, 20 mM, pH 6.5), stirred, and then incubated at 40 °C for 1 h with purified lichenase enzyme (200 µL, 10 U). After the addition of 5 mL of 200 mM sodium acetate buffer pH 4.0 and centrifugation at 5000 rpm for 10 min, an aliquot (100 µL) was then hydrolyzed to completion with purified β-glucosidase (100 µL, 0.2 U) during incubation at 50 °C for 10 min. A total of 50 mM acetate buffer pH 4.0 was used as the reaction blank. The D-glucose produced was assayed using a glucose oxidase/peroxidase reagent (GOPOD) (3 mL). The absorbance was measured at 510 nm (Agilent 8453 UV-visible spectroscopy system, Agilent Technologies, Inc, Santa Clara, CA, USA). The *β*-glucan content was expressed as a percentage of d.m.

#### 2.3.5. Analysis of Resistant Starch

The resistant starch content was determined according to the Megazyme protocol [39]. Summarily, 4.0 mL of pancreatic α-amylase (10 mg/mL) containing amyloglucosidase (AMG) (3 U/mL) are added to 100 mg oat samples and incubated for 16 h at 37 °C in a shaking water bath. During this time, the combined action of the two enzymes hydrolyzes non-resistant starch to D-glucose. Adding 4.0 mL of 99% ethanol stops the reaction, and centrifugation recovers the RS as a pellet. After that, the pellets are centrifuged twice (1.500× *g*, 10 min) and washed with 2 mL of ethanol (50% *v/v*). Decantation is used to eliminate free liquid. RS in the pellet is dissolved in 2 mL of 2 M KOH by agitating the mixture strongly over a magnetic stirrer in an ice-water bath. AMG is used to quantitatively hydrolyze the starch to glucose after neutralization with 8 mL of 1.2 M (pH 3.8) acetate buffer. GOPOD reagent (3 mL) is used to measure D-glucose, which is a marker for the sample’s RS concentration. By combining the original supernatant and the washings, increasing the volume to 100 mL, and using GOPOD to measure the D-glucose level, non-resistant starch can be identified. The absorbance of each solution was measured at 510 nm against the reagent blank. The resistant starch content of the samples was calculated by using the Megazyme Mega-Calc™ spreadsheet, and the results were expressed as a percentage per dry matter (d.m.).

#### 2.3.6. Extraction of Total Phenolic Compounds

Alkaline hydrolysis was applied at room temperature for 4 min using 10 mL of 4 M NaOH to release phenolic compounds from 500 mg of OF and ground OHs. From the hydrolyzate, phenolic compounds were extracted according to the procedure described by Žilić et al. [40]. The extracts were evaporated to dryness under the N_2_ stream at 25 °C (Reacti-Therm nitrogen evaporator system 18821, Thermo Fisher Scientific Inc., Waltham, MA, USA) and the residues were redissolved in methanol. Such prepared methanolic solutions were used for the analyses of total phenolic compounds and phenolic acids. All the extractions were performed in triplicate for each sample and kept at −70 °C before analyses.

#### 2.3.7. Analysis of Total Phenolic Compounds (TPCs)

The total phenolic content was determined by the Folin-Ciocalteu assay as described by Singleton et al. [41]. Briefly, 300 and 100 μL of the OF extracts and OH extracts were transferred into test tubes, and their volume was filled up to 500 μL with distilled water. After the addition of Folin-Ciocalteu reagent (250 μL) and 20% aqueous Na_2_CO_3_ solution (1.25 mL), the tubes were vortexed and the absorbance of the mixture was measured at 750 nm after 40 min and centrifugation at 8000 rpm. The content of total phenolics was expressed as µg of gallic acid equivalent (GAE) per g of dry matter (d.m.).

#### 2.3.8. Analysis of Phenolic Acids

To determine the phenolic acids, the clear supernatants were filtered through the 0.45 μm nylon filter, and the pure extracts were analyzed using the HPLC-DAD system (Thermo Scientific Ultimate 3000). The chromatographic separation was performed on the Thermo Scientific Hypersil GOLD aQ C18 column (150 mm × 4.6 mm, i.d., 3 µm) at 25 °C using a gradient mixture of 1% formic acid in water (solvent A) and 100% methanol (solvent B) at a flow rate of 0.8 mL/min and run time of 30 min. The solvent gradient was programmed as previously described by Žilić et al. [42]. The chromatograms were recorded at 280 nm by monitoring the spectra within the wavelength range of 190–400 nm. Standards of gallic acid, 3,4-dihydroxybenzoic acid, chlorogenic acid, vanillic acid, caffeic acid, syringic acid, *p*-coumaric acid, sinapic acid, ferulic acid, and isoferulic acid were used (10, 20, 40, 50 and 100 µg/g). The identified phenolic acids’ peaks were confirmed and quantified by data acquisition and spectral evaluation using the Thermo Scientific Dionex Chromeleon 7.2. Chromatography software. The content of phenolic acids was expressed as µg per g of d.m.

#### 2.3.9. Analysis of Total Antioxidant Capacity (TAC)

The antioxidant capacities of OH fine powder and OF were measured according to the QUENCHER method described by Serpen et al. [43], using a 7 mM aqueous solution of ABTS (2,2-azino-bis/3-ethil-benothiazoline-6-sulphonic acid) with 2.45 mM K_2_O_8_S_2_ as the stock solution. The working solution of ABTS^•+^ was obtained by diluting the stock solution in water/ethanol (50:50, *v/v*). Depending on the sample, 2 and 9 mg of ground OHs and OF, respectively, were mixed with 20 mL of ABTS^•+^ working solution, and the mixture was rigorously shaken for 25 min. Afterwards, the centrifugation absorbance was measured at 734 nm. The total antioxidant capacity was expressed as the Trolox equivalent antioxidant capacity (TEAC) in mmol of Trolox per kg of d.m.

### 2.4. Baking Functionality Properties

#### 2.4.1. Gelling Properties

The water solubility index (WSI), water absorption index (WAI), and swelling power (SP) were determined according to the method described by Cornejo and Rosell [44]. Powdered OHs and OF (1 g) were weighed into centrifuge tubes and 20 mL of distilled water was added. The tubes were shaken in a water bath for 15 min at 90 °C and then centrifuged at 3000× *g* at 4 °C for 10 min. The WAI was calculated as the quotient of the mass of the sediment and the initial mass of the sample. Dry matter of the supernatant evaporated in the ventilation oven for 12 h at 110 °C was used for the WSI calculation. The SP was calculated as the quotient of the sediment mass and the difference between the initial mass of the sample and the dry mass of the evaporated supernatant. Each sample was analyzed in three replicates, and the WSI and SP values were expressed in g.

#### 2.4.2. Solvent Retention Capacity (SRC)

Solvent retention capacity (SRC), a test that indicates the ability of flour to retain individual diagnostic solvents (distilled water, 50% sucrose, 5% sodium carbonate, and 5% lactic acid water solutions) based on the swelling behavior of polymer networks in flour, was determined according to the American Association of Cereal Chemists (AACC) method 56–11 adapted by Haynes et al. [45]. Each flour polymer network is associated with the corresponding diagnostic solvent. Considering the analyzed samples, OH powder and OF, two solvents were individually used to determine the SRC values: 50% sucrose in water and 5% lactic acid in water. Five grams of sample (OH powder and OF) were weighed and vortexed in 25 mL of an appropriate solvent for a total of 25 min at one-minute intervals every 5 min to allow the samples to solvate and swell. After centrifugation at 3000 rpm for 10 min, the SRC values were calculated and expressed as percentages of the mass of flour gel after exposure to the solvent in relation to the original flour weight.

### 2.5. In Vitro Multistep Enzymatic Digestion Protocol

To determine the potential OF and ground OH digestibility for human consumption an in vitro multistep digestion procedure was applied. The method, consisting of oral, gastric, duodenal, and colon phases, was proposed by Papillo et al. [46] and modified according to Hamzalioğlu and Gökmen [47]. Digestion fluids simulating the saliva (simulated salivary fluid, SSF), gastric juice (simulated gastric fluid, SGF), and duodenal juice (simulated duodenal fluid, SDF), as well as pepsin solution in 0.1 M HCl, bile salts, pancreatin solution in distilled water, protease water solution, and viscozyme L, were used and prepared according to the procedure described by Hamzalioğlu and Gökmen [47]. The multistep digestion conditions are described in detail by the same authors. Samples obtained after in vitro digestion were filtered through qualitative filter paper, air-dried in a ventilated oven (Memmert UF 55; Memmert GmbH + Co. KG) for 2 h, and then dried to constant mass at 105 °C for 4 h. After weighing the samples, their digestibility was calculated according to the following equation:Digestibility = ((m_0_ − m_d_)/m_0_) × 100(1)
where m_0_ is the mass of the absolutely dry sample prior to digestion, and m_d_ is the remaining (undigested) mass of the absolutely dry sample. The digestibility of dry matter was expressed as a percentage.

### 2.6. Antimicrobial Activity Evaluation

#### 2.6.1. Extract Preparation

The samples were prepared according to the recipe from the traditional medicine of Serbia [9]. Briefly, OF and powdered OH samples (10 g) were poured into 150 mL of water and boiled until the volume was reduced to 100 mL. Samples were filtered and the supernatant was left to cool. Afterward, the extract prepared after alkaline hydrolysis, as described by Žilić et al. [40] and evaporated to dryness under the N_2_ stream at 25 °C, was dissolved in water and used for the antimicrobial assay.

#### 2.6.2. Antimicrobial Assay

The antimicrobial effect of oat samples was tested on six important pathogens of the gastrointestinal tract, namely: *Listeria monocytogenes, Staphylococcus haemolyticus, Salmonella typhimurium, Enterococcus faecalis, Escherichia coli*, and *Shigella flexneri*. An overnight culture of bacteria was produced by inoculating several colonies in MH (Muller Hinton) broth and incubating in a thermostat at 37 °C for up to 24 h. The cultures thus prepared were brought to a cell concentration of 1 × 10^8^ CFU/mL with a McFarland tube densitometer (DEN-1, BIOSAN, Riga, Latvia). The densitometers are designed and factory-calibrated to measure the turbidity of cell suspensions in a variety of life science applications. Dilutions were made from the obtained bacterial suspensions so that the final concentration of bacteria was 10^6^ CFU/mL. First, an attempt was made to determine the MIC (minimum inhibitory concentration) values of the tested water extracts in microtiter dilution plates with 96 wells. As the wells in the plates were defined to hold 200 mL of total volume (test substance, liquid broth, and bacterial suspension), and water extracts for inhibiting the growth of the tested bacteria required higher concentrations, we used a modified method. Different volumes of tested extracts were added to Petri dishes with 1 mL of overnight culture of tested bacteria with a final concentration of 10^6^ CFU/mL. A medium containing the MH agar with a melting point below 45 °C was poured over it. Petri dishes prepared this way were incubated in a thermostat at 37 °C for 48–72 h after hardening the agar. Each concentration was determined in triplicate. After incubation, grown colonies were counted and compared with control Petri dishes in which only bacterial suspensions and medium for bacterial growth were added.

### 2.7. Statistical Analysis

The data were reported as a mean ± standard deviation of three independent repetitions per sample. Statistical analyses were performed using Minitab19 Statistical Software. The one-way ANOVA analysis of variance with Tukey’s test was. Pearson’s correlation coefficients (r) were calculated to evaluate the relationship between the individual parameters. Differences between the means with probability *p* < 0.05 were accepted as statistically significant.

## 3. Results and Discussion

### 3.1. Nutritional Profile of OF and OH

#### 3.1.1. Basic Chemical Composition

The results of the basic chemical composition of the investigated OF samples and powdered OHs are shown in Table 1.

The starch content in the OF of all three oat genotypes was expectedly high, ranging from 55.71% in the yellow oat genotype to 57.58% in the black oat genotype. On the other hand, starch was not determined in the OHs of all three genotypes. Furthermore, oat starch is the main grain ingredient predominant in the endosperm, and differs in content between 51 and 65% due to variations caused by environmental factors during cultivation and plant genotype [48]. Oat starch exhibits several distinct structural characteristics in contrast to other cereal starches, including a clustered granular structure, smaller granule size, lower relative crystallinity, and higher concentration of amylose-lipid complexes [17]. These properties allow starch to be used as a food ingredient or additive for the improvement of texture, emulsion stability, and moisture retention. Oats are suitable for human consumption and have various applications, including oat flour and oatmeal, biscuits, noodles, bread, bars, and yogurt [49].

The highest fat content determined in our samples (7.20%) was in the brown OF, and while it amounted to around 1% in the OHs it did not differ significantly among the oat genotypes. Oats are a unique cereal, with 2–18% fat, which accumulates mainly in the endosperm, unlike other oily seeds which typically store fats in the embryo [50]. Oat lipids consist of 23% saturated (mostly palmitic acid), 34% monounsaturated (primarily oleic acid), and 43% polyunsaturated (primarily linoleic acid) fats [51]. This composition affects food oxidation, where the amount of unsaturated fatty acids is especially significant, and hence the flavor of oat products. The results of our study follow those of Ibrahim et al. [52], who reported fat contents ranging from 6.16% to 6.67%, and ash contents varying between 3.90% and 6.02% among the five oat cultivars cultivated in Pakistan. The ash content determined in our study was the highest in yellow OHs (5.85%) and the lowest in yellow OF (2.19%). According to Usman et al. [53], oat grains had ash and fat contents of 3.7% and 4.5%, respectively. Neitzel et al. [54] reported that OHs contained 6.27% ash. Zhu et al. [55] found that whole oats comprised 14.88% crude protein, 8.16% crude fat, 1.57% ash, and 60.15% total starch. Varietal variances, climatic circumstances, soil composition, and cultural practices could all represent contributing factors to the discrepancies in the study’s conclusions [56].

After removing the OHs, the remainder of the oat grain—groat generally contains 15–25% protein. The protein content of oat grains rises from the center to the outside [57]. The starchy endosperm has about 12% protein, while the bran (pericarp, testa, nucellus, aleurone, and some subaleurone) has 18–26% and the germ has 29–38% [58]. The total protein content detected in our study was higher in the investigated OF, ranging from 13.43% in black oats to 17.38% in yellow oats, than in OHs where it varied between 3.09% in black oats and 4.79% in yellow oats. Ibrahim et al. [52] reported protein contents ranging from 8.13% to 12.69% among the five oat cultivars cultivated in Pakistan, while Usman et al. [53], reported average protein levels of 13.5% in oat grains. Zhou et al. [23] reported 5.3% of total protein in the ground OHs. Kouřimská et al. [51] analyzed hulled, dehulled, and nude oat grains (without OHs) and found the range of protein content was between 16.75 and 17.78% (d.m. basis), while the fat content was between 3.16% and 5.82%. Oat grain is the only cereal crop that contains avenalin, salt-soluble legumin-like globulin nutritional quality to soybean proteins, as the major storage protein (70–80%) [6,58]. In contrast to the predominant alcohol-soluble prolamins, commonly referred to as gluten proteins, which comprise 60–80% of the total protein found in the *Triticeae* cereals: wheat (gliadins and glutenins), barley (hordeins), and rye (secalins) avenins make up about 10–15% of the total protein in oats [59,60,61]. As long as there is no cross-contamination with other gluten-containing cereals and the food’s gluten content is less than 20 mg/kg, oats, oat products can be regarded as gluten-free under EU law ((EU) No 828/2014) [62]. The content of alcohol-soluble avenin prolamin detected in our study ranged from 11.73% (yellow oats) to 13.86% (brown oats) in the OF, and from 13.36% (yellow oats) to 18.12% (black oats) of total protein found in the oat OHs. In terms of percentage of dry matter, these values are well below the benchmark of 20 mg/kg of oats, which means that the OF and OHs are safe for human consumption as gluten-free. To sum up, the primary distinctions between the prolamins of different cereals are found in their physicochemical properties, including their source, solubility, molecular weight, disulfide bonds, and amino acid makeup [63]. Despite a lack of research on avenin, the nutritional qualities of oats and the health benefits of avenins have received more attention in recent years. Compared to other prolamins, avenin is more hydrophilic; its maximum solubility was found in 45% (*w/w*) ethanol [61].

#### 3.1.2. Dietary Fibers

Dietary fibers are crucial nutrients that possess beneficial properties such as regulating various physiological processes, ranging from bowl regulation to treating chronic illnesses [52]. According to Zhu [64], the health benefits of dietary fibers include antioxidative properties, possible anticancer effects, control of body weight and glycemic levels, neuroprotective qualities, protection of retinal health, hypolipidemic effects, hepatoprotective properties, and potential anti-aging effects. Some studies indicate that the antioxidant effect of dietary fiber is the result of the phenolic compounds bound to polysaccharide complexes that comprise the dietary fiber [65].

The results of the investigated dietary fibers are presented in Table 2.

Significant differences in the fiber composition of the studied oat samples were detected, namely between the powdered OH which had higher lignocellulosic fibers, NDF (80.27–83.82%), ADF (36.04–38.96%), ADL (1.86–6.00%), hemicellulose (41.31–46.11%) and cellulose (32.96–35.32%) contents, compared to those in OF (15.83–32.36%, 3.11–3.65%, 0.89–1.29%, 12.72–28.71%, 2.22–2.45%, respectively), which can be explained by the fact that OHs consist mainly of lignocellulosic fiber.

These results are in accordance with the previous research of de Oliveira et al. [66] where OHs contained 40.1% cellulose, 25.1% hemicellulose, 26.1% lignin, and 8.7% ash. Furthermore, Neitzel et al. [54] reported that OHs contained 66.19% holocellulose, 29.80% α-cellulose, and 25.44% lignin. Neitzel et al. [54] reported that OHs contained 66.19% holocellulose (total polysaccharide fraction after removal of extractives and lignin), 29.80% α-cellulose, and 25.44% lignin. A previous study by Žilić et al. [67] found that hull-less oat grain contained on average 13.11% hemicellulose, 1.41% cellulose, 15.16% NDF, 2.04% ADF, and 0.98% lignin, which follows the results of our study. Cell wall components such as cellulose, hemicellulose, lignin, and silica make up the NDF fraction. Since lignin is completely indigestible and its presence decreases the availability of the plant material’s cellulose and hemicellulose components, it is significant from a nutritional aspect [68]. Because OH contains considerable amounts of fiber, most research on OH valorization has been on using lignocellulose for the manufacture of biofuel or animal feed [23].

However, recent studies have demonstrated that OHs can find different applications in the food industry. For example, given that micronized OHs are a food byproduct rich in dietary fiber and polyphenols with antioxidant qualities, they can be used as an additive in bread making to improve the nutritional and textural properties of gluten-free bread [69].

The OF samples had a higher *β*-glucan content (from 4.07% to 5.33%) than those in OHs (from 0.03 to 0.06%). Majumdar et al. [70] reported levels of *β*-glucan in three oat lines ranging from 4.43% to 6.46%, which is in accordance with our results. Oat *β*-glucans are soluble dietary fibers that can be consumed by gut microbiota in the colon, leading to the production of short-chain fatty acid (SCFA) metabolites [71]. The European Commission approved a health claim for oat *β*-glucan in 2011 based on the European Food Safety Authority’s scientific conclusion that consuming oats lowers postprandial glycemia [72].

Resistant starch is not digested in the small intestine but can instead be utilized in the colon by gut microbiota [73]. Resistant starch in oats, accounting for 29.31% of the starch content in raw granular form, can modulate blood glucose and contribute to the food glycemic index value [71]. The resistant starch in OFs ranged between 2.43% in brown oat grains and 2.95% in yellow oat grains, while its presence was not detected in the OH samples. However, Xia et al. [74] reported a slightly higher content in the untreated whole-grain oats, which amounted to 5.31%. According to Zhu et al. [55], whole oats samples comprised 4.72% resistant starch, and 13.53% total dietary fiber, while 5.08% of the whole-grain oats were *β*-glucan. 

### 3.2. Phenolic Compounds and Antioxidant Capacity

The contents of phenolic compounds, phenolic acids, and antioxidant capacities of the investigated oat samples are given in Table 3. According to our study, the content of total phenolic compounds determined in the OF samples of the oat genotypes grown in Serbia was close to the content measured in grains of oat genotypes grown in the Czech Republic, as reported by Alemayehu et al. [75]. These authors studied the content of total phenolics in dehulled oat grains of genotypes originating from nine world countries and reported that the content was the highest (1688.0–2016.0 μg GAE/g) in the grains of oat genotypes originating from India. In addition, according to the results obtained in our study, OHs had 13 to 25 times higher contents of total insoluble-bound phenolic compounds (12,086.76 to 24,352.48 μg GAE/g d.m.) compared to OF (841.89–982.08 μg GAE/g d.m.). A previous study by Žilić et al. [21] reported a low content of soluble free phenolic compounds in the grain of the four analyzed standard yellow-colored hull-less oat genotypes. Varga et al. [76] also reported that OHs of twenty differently colored oat genotypes had higher both soluble free and insoluble-bound phenolic compounds than groats, as well as that the amount of bound phenolics was ten times higher than that of free phenolics. On the other hand, Emmons et al. [77] reported that the average content of total phenolic compounds in OF and OHs was not significantly different. There have been reports of various cereal species hierarchies based on their levels of total phenolic and antioxidant activity [21,78]. In addition to genotype, the antioxidant properties were found to be influenced by location and genotype x location interactions [76]. Conversely, Emmons et al. [79] reported that there were no significant differences in the content of total free phenolic compounds, *p*-coumaric, and ferulic acid in three oat genotypes grown in seven different locations, unlike the total antioxidant capacity and content of other phenolic compounds, which differed significantly. Five phenolic acids were detected in oat samples, *p*-coumaric, ferulic, isoferulic, vanillic, and syringic acid. In the study by Žilić et al. [21], a high content of soluble free caffeic acid was measured in the grains of hull-less oat genotypes. According to the findings of Varga et al. [76], the OHs and grain of the tested genotypes contained a total of 28 soluble phenolics, including phenolic aldehydes, benzoic acids, hydroxycinnamic acids, mono- and dihydroxycinnamoyl glycerol esters, and avenanthramides. The most prevalent phenolic aldehyde in the OHs and grain reported by Emmons et al. [77] was vanillin. In our study, the content of vanillic acid, an oxidized form of vanillin, ranged from 16.66 μg/g in black OF to 23.05 μg/g in yellow OF, and from 101.44 μg/g in yellow OHs to 619.38 μg/g in black OHs. In general, the phenolic acids content was higher in the OHs, especially of the brown and black genotypes, except for the *p*-coumaric acid, where the content was significantly higher in the yellow oat genotype’s OHs. The *p*-coumaric acid content in the OHs was about 74, 34, and 33 times higher in the yellow, brown, and black OHs than in the OF, respectively. Ferulic acid was predominant, in both the OF (395.88 to 589.14 μg/g d.m.) and OHs (4987.02 to 13,794.82 μg/g d.m.). Ferulic acid is one of the phenolic acids that play a crucial part in anti-inflammatory processes through cyclooxygenase inhibition, both tumor necrosis factor-alpha and prostaglandin E2, as shown by the improved arterial endothelial function seen in both in vitro [80] and in vivo. Additionally, ferulic acid possesses gut-modulating qualities supporting its overall ability to promote health [26]. Compared to *p*-coumaric acid, especially ferulic acid, the content of isoferulic acid and syringic acid was low in all the tested samples.

The antioxidant capacity of the samples investigated in our study was higher in OHs, ranging from 42.31 mmol Trolox/kg d.m. in yellow OHs to 53.16 mmol Trolox/kg d.m. in brown OHs, and from 22.61 mmol Trolox/kg d.m. in black OF to 25.06 mmol Trolox/kg d.m. in brown OF (Table 3). The total antioxidant capacities of oat grains determined by different methods and expressed in Trolox equivalents reported by various sources varied between 1.7–3.0 μmol Trolox/g and 32.9–117.9 μmol Trolox/g [75]. Furthermore, in our study, unlike the phenolic compounds, the antioxidant capacity of OHs was 1.6 to 2.5 times higher compared to that of OF. This may indicate the antagonistic effect of the compounds present in the oat.

### 3.3. Baking Functionality

The properties of oat OF and powdered OHs that impact baking functionality, i.e., gelling properties WSI (water solubility index), WAI (water absorption index), SP (swelling power), and solvent retention capacities (SRC), are presented in Figure 1.

The WAI, WSI, and SP varied significantly. The WAI values were respectively lower in OHs than in OF. Except for the yellow OF, which had a lower WSI than the OHs, brown and black OF had a higher WSI than the respective OH. The SP was higher in all the OF than in the OHs. However, all of these gelling properties can be considered exceptional, given that the starting material before water treatment weighed 1 g, and the results indicate that the starting material absorbed 3–5 times more water than its initial mass. A significant correlation between the WAI and starch content (r = 0.646**) can be attributed to the fact that the degradation of the crystalline structure of the starch and the increase in amylose content leaching represent the cause of the maximum water absorption capacity. According to Ibrahim et al. [52], the OF with a higher percentage of polysaccharides has a far higher potential to absorb water. Compared to wheat flour, OF has a better water absorption ratio and dough-softening qualities because of its increased *β*-glucan content [81]. A positive correlation between the WAI and total protein content (r = 0.591*) follows the statement from the research of Ibrahim et al. [52] that the flour’s ability to absorb water is determined by its polar amino acid content. The WAI showed positive correlations with *β*-glucan (r = 0.556*), and resistant starch contents (r = 0.588*), as well as a negative correlation with the NDF content (r = −0.587*). Furthermore, Ibrahim et al. [52] reported that the five investigated cultivars’ water absorption capacities varied greatly, ranging from 173.66 to 188.33%, while Smuda et al. [82] reported a slightly lower water absorption index of oat bran of 115.2%. Varietal differences, variations in soil composition, variations in climate, and variations in agronomic techniques could be the reason why the cultivars differ considerably in terms of water absorption capacity in the study by Ibrahim 2020 [52].

The swelling power manifested positive correlations with the protein content (0.769**), *β*-glucan (0.695**), starch (0.705**), and resistant starch contents (0.691**), and a strong negative correlation with the NDF fiber content (r = 0.718**). According to a previous study on wheat genotypes conducted by Nikolić et al. [83], the presence of fiber-rich bran in whole-wheat flours had a significant impact on their baking functionality. For instance, sucrose SRC had the highest correlation with the lignin content, whereas the starch content improved absorption capacities for 50% sucrose and 5% lactic acid, respectively.

Since they are a rich source of dietary fiber with high water absorption capacity, OHs are considered a promising component for gluten-free baking [66]. Furthermore, thanks to their high water absorption capacity, utilizing the cellulose fibers from OHs to produce hydrogel and cellulose nanocrystals for food packaging is encouraging because, in addition to the recycling of the cellulose fibers and by-products of grain processing, they can be used to create valuable products that offer distinct attributes [66].

SRC tests have become increasingly popular for assessing the quality of wheat products by analyzing the functionality of flour with four diagnostic SRC solvents. However, their application for the evaluation of oat flour has been scarce until recently. The results of our study showed that the sucrose SRC ranged from 181.06% in black OF to 203.00% in brown OHs, while the lactic acid SRC ranged from 173.88% in black OF to 204.78% in brown OHs. In a study in which 30 oat varieties were analyzed, Zhang et al. [32] found that lactic acid SRC values varied between 76.20% and 154.00% and for sucrose SRC between 68.35% and 90.25%. They also reported that there were correlations between the *β*-glucan content and lactic acid SRC (0.373*), as well as between the ash content and lactic acid SRC (0.481*), while sucrose SRC did not correlate to any of the OF components. Conversely, the results of our study indicate that the lactic acid showed no significant correlations to the chemical constituents of the oat grain and OH. However, the sucrose SRC was negatively correlated with the starch (r = −0.529*) and resistant starch content (r = −0.511*), *β*-glucan (r = −0.493*), and total protein (r = −0.467*) content, and positively correlated with the NDF fiber content (r = 0.483*). The discrepancies between the correlations determined by different authors can be attributed to the genotypic differences, as well as the number of samples used in the study. Lactic acid is considered a measure of gluten strength, and is associated with protein quality, and therefore linked to flour’s capacity to absorb water [32]. Research by Hammed et al. [84], and Pasha et al. [85] showed that the sucrose SRC ranged from 68.35 to 90.25% and averaged 82.49%, which was less than that of wheat flours (121.9–140.6% and 125.0–163.0%, respectively). The explanation could be that the pentosan content of oats is substantially lower than that of wheat, and sucrose SRC is correlated with pentosan content [32].

### 3.4. In Vitro Digestibility

The results of the multistep in vitro enzymatic digestibility protocol utilized to determine the potential digestibility of the OF and powdered OH samples are depicted in Figure 2. 

The digestive properties of foods derived from oats have been receiving a lot of attention lately. Research suggests that oats, which are inherently slow-digesting cereals, can help control postprandial blood glucose levels with consistent daily consumption [71]. It is evident from the results depicted in Figure 2 that the OF samples had significantly higher digestibility (from 44.54% to 48.24% in yellow and brown OF, respectively) than the powdered OH samples (from 12.02% to 16.69% in black and brown OH, respectively). Lower digestibility was anticipated because of the OH′s significantly higher fiber content, as supported by the strong negative correlation (r = −0.966**) between digestibility and NDF content. Furthermore, Schefer et al. [86] reported that phenolic compounds interact with proteins, lipids, and carbohydrate constituents, which further inhibit the digestive enzymes via chemical interactions, causing them to precipitate and reduce their activity in the degradation of carbohydrates. A strong negative correlation between the digestibility and total phenolic compounds determined in our study (r = −0.943) is in accordance with the abovementioned statement. On the other hand, Zhang et al. [87] found that bound phenolics from rice bran dietary fiber were released during simulated gastrointestinal digestion. However, over a quarter were released during colonic fermentation by microbiota stimulating the growth of certain beneficial bacteria during colonic fermentation.

Oat *β*-glucans have a high viscosity, which slows gastric emptying and delays food digestion, reducing glucose release due to starch hydrolysis. This increases chyme viscosity, preventing stomach emptying and affecting glucose transfer to enterocytes [71]. The hypoglycemic effect of *β*-glucans begins in the stomach’s stage of food digestion, where they form extremely viscous solutions in an acidic environment. This slows the digestion of carbohydrates and gastric contents into the duodenum, lowering glucose release and reducing postprandial glycemia and insulin requirement [88]. In our study, a significant negative correlation between the digestibility and the *β*-glucan (r = −0.869**) supports these statements.

### 3.5. Antimicrobial Activity of the Oat Extracts

Traditional Serbian medicine uses oat decoction to treat various digestive tract issues, including diarrhea, pancreatitis, and excess stomach acid [15]. The abovementioned conditions could represent symptoms or possible complications of bacterial infections. For this reason, the antimicrobial effect of the oat extract samples was investigated on six important pathogens of the gastrointestinal tract (*Listeria monocytogenes*, *Staphylococcus haemolyticus*, *Salmonella typhimurium*, *Enterococcus faecalis*, *Escherichia coli*, and *Shigella flexneri*). Extracts obtained after hydrolysis and samples prepared according to the traditional recipe were tested (Table 4).

The samples obtained by hydrolysis showed a higher bactericidal activity, which can be associated with a higher content of polyphenols. The extracts obtained from the yellow OF and OHs showed the highest antimicrobial activity against the majority of the investigated bacterial strains. The most sensitive bacteria were *L. monocytogenes* and *E. coli*, which were suppressed by almost all samples in amounts less than 200 µL/mL. Moreover, OF and OH extracts of the yellow oat samples obtained after hydrolysis were also active in inhibiting the growth of *E. faecalis* (140 µL/mL for both samples) and *Shigella flexneri* (150 and 160 µL/mL for OH and OF samples, respectively). The least sensitive to the activity of all extracts obtained by hydrolyzation was *S. typhimurium* (Table 4).

As for the samples obtained by the traditional recipe, the strongest bactericidal activity was manifested by brown OF extract and black OH extract, especially against *L. monocytogenes*, *E. coli*, and *S. haemolyticus*. As in the case of hydrolyzed samples, *S. typhimurium* was the least sensitive bacterial strain (Table 4).

Alrahmany et al. [89] concluded that there is a direct connection between antimicrobial activity and the presence of phenolics in the bran extracts by showing that oat brans treated with enzymes cellulase, amyloglucosidase, and viscozyme reduced the growth of *E. coli* by 37% compared to controls. A study reported that *Escherichia coli* was susceptible to the antimicrobial action of powdered sprouted oats extracts containing phenolic acids and avenanthramides [90].

Phenolic acids are among the most abundant phenolic components in oats, and the most common among them are cinnamic acid and benzoic acid derivatives [77]. Phenolic acids are mostly bound to proteins, sugars, or fatty acids in oat grains [91]. Chen et al. [92] demonstrated that cellulase treatment increased the availability of the majority of the phenolic compounds from oat bran, while heating-only treatment had no significant influence. On the other hand, heat treatment can cause incorporation of phenolic acids and other polyphenolic compounds into high molecular weight polymers called melanoidins, which are formed as a result of Maillard reaction [93]. In our study, despite the fact that the content of phenolic acids in traditionally prepared extracts was significantly lower than in the hydrolyzed ones, their antimicrobial effect was almost comparable for the majority of the studied bacterial strains. This indicated that some other compounds, in addition to phenolic acids, contributed to the demonstrated effects. Previous studies suggested that Maillard reaction products such as melanoidins also possess antimicrobial effects [94,95]. Kukuminato et al. [94] showed that the antibacterial effect of some melanoidins is comparable to that of nisin solution in a concentration 350 IU/mL. Rurian-Henares and Morales [96] pointed out that water-soluble melanoidins eliminated pathogenic bacteria strains (*E. coli*) by generating irreversible changes in both the inner and outer membranes.

The antimicrobial activity of phenolic acids is presented in a number of studies and is connected with their ability to interact with cell membranes or to cross them. Campos et al. [97] showed that phenolics could damage cell membranes causing the efflux of cellular components. Furthermore, some studies reported that phenolic acids can cross cell membranes of *E. coli* and exert their antimicrobial activity inside the cytoplasm [98,99].

The best antimicrobial activity of yellow OH extract, obtained by hydrolysis, can be explained by the high share of phenolic acids, among all the analyzed oat samples, for which strong antimicrobial activity has been proven in numerous earlier studies. Among them, *p*-coumaric acid, ferulic acid, caffeic acid, and vanillic acid are the most prevalent (Table 4). The high contennt of ferulic, *p*-coumaric and vanillic acids was also reflected in the good antimicrobial activity of black and brown OHs, prepared according to the traditional recipe.

Many studies have proven the high antimicrobial activity of phenolic acids concerning bacteria that cause food spoilage and gastrointestinal problems. Borges et al. [100] found that ferulic acid had antimicrobial activity against *Escherichia coli*, *Listeria monocytogenes*, *Staphylococcus aureus,* and *Pseudomonas aeruginosa*. The results suggest that ferulic acid compromises the integrity of the cytoplasmic membrane and leads to irreversible changes in membrane properties (charge, intra and extracellular permeability, and physicochemical properties) through hydrophobicity changes, a decrease in the negative surface charge, and occurrence of local rupture or pore formation in the cell membranes, with consequent leakage of essential intracellular constituents. It is possible to observe that the percentage of cells with membrane damage considerably increased with the phenolic acid concentration. As ferulic acid, along with *p*-coumaric acid, is the most abundant in the most active OH samples in our tests, it can be concluded that its synergistic activity, along with other phenolic components, contributed to its good antimicrobial activity.

The *p*-coumaric acid, the most abundant in black and brown OHs, slightly less in the OHs of the yellow oat variety, has shown considerable antimicrobial activity through a dual mechanism of action. The results of Lou et al. [101] showed that *p*-coumaric acid effectively inhibited the growth of all test bacterial pathogens, and the MIC values ranged from 10 to 80 μg/mL. The *p*-coumaric acid was tested for antibacterial potential against Gram-positive (*S. pneumoniae* ATCC49619, *B. subtilis* 9372, *S. aureus* 6538) and Gram-negative bacteria (*S. dysenteriae* 51302, *E. coli* ATCC25922, *Salmonella typhimurium* 50013). These studies further showed that *p*-coumaric acid significantly increased the outer and plasma membrane permeability, resulting in the loss of the barrier function. The results demonstrate that *p*-coumaric acid has dual mechanisms of bactericidal activity: disrupting bacterial cell membranes and binding to bacterial genomic DNA to inhibit cellular functions, ultimately leading to cell death (apoptosis). Ojha and Patil [102] revealed that *p*-coumaric acid has the ability to inhibit the DNA repair mechanisms of *L. monocytogenes* by in vivo inhibition of its inducer—RecA expresion The results obtained in our study are in correlation with the research presented because the highest inhibitory activity on the growth of pathogenic bacteria was shown by OH samples in which *p*-coumaric acid, along with ferulic acid, is the most abundant.

Very important information was provided by the research, in which it was proven that the addition of *p*-coumaic acid to a probiotic may enhance the probiotic properties of bacteria such as *Lactobacillus acidophilus* LA-5 and *Lacticaseibacillus rhamnosus* GG [103]. It was also confirmed that this phenolic acid has a protective effect on colon pretumors induced by 1,2 dimethylhydrazine (DMH) in rats [104].

Moreover, many studies have proven the distinct antimicrobial activity of other phenolic acids that are slightly less abundant in our samples than the first two described, such as vanillic acid, caffeic acid, and 3,4-dihydrohybenzoic acid. Syafni et al. [105] have shown significant and dose-dependent inhibition activity of extracts and fractions of leaves, stems, and roots of *T. cinense* L. and their dominant compounds 3,4-dihydrohybenzoic acid and 3,4-dihydrohybenzaldehyde toward *E. coli* 25922, *Staphylococcus aureus* ATCC 25923, and *Vibrio cholera* Inaba. The experiments also revealed that the extracts and isolated 3,4-dihydrohybenzoic acid and 3,4-dihydrohybenzaldehyde are also active against *Salmonella thypimurium* ATCC 14028 NCTCC 12023, which caused not only diarrhea but also salmonellosis with fever and abdominal cramps as additional symptoms. These results support the traditional use of this plant and its two dominant components in treatment of diarrhea. Although 3,4-dihydrobenzoic acid was present only in traditionally prepared samples, the abovementioned results from previous studies support our outcomes in terms of the possible application of oat OHs in the treatment of gastrointestinal issues induced by, amongst others, bacterial infections. 

Previous studies have shown that vanillic acid, the most abundant in black and brown OHs, and OHs of the yellow oat variety, has proved its efficacy as an antimicrobial compound against foodborne bacteria such as *Staphylococcus aureus*, *Escherichia coli*, and carbapenem-resistant *Enterobacter cloacae*, *Enterobacter homaechei*, *Sarcina* spp., and *Candida albicans* [106]. The existing literature indicates that vanillic acid can disrupt the biofilm formation of clinical pathogens. Additionally, disruption of cell membrane integrity by vanillic acid may result in the formation of pores and leakage of intracellular components into the extracellular space [107]. Moreover, Ingole et al. [108] indicated that vanillic acid could help patients with ulcerative colitis caused by dextran sulfate sodium (DSS), by reducing the severity of clinical symptoms, which justifies its use as a flavoring agent.

Prior investigations have demonstrated that the antimicrobial activity of the plant extracts is frequently based on the synergistic activity of both dominant and the less abundant compounds. Ijabadeniyi et al. [109] evaluated the antimicrobial effects of two phenolic acids (caffeic and ferulic) against foodborne pathogens, *Escherichia coli* O157:H7 ATCC 43888 and *Listeria monocytogenes* ATCC 7644, in ready-to-eat meats that are susceptible to pathogenic contamination during their production, distribution, and sale. These results indicate the ability of caffeic and ferulic acids, individually and in combination, to reduce pathogenic contamination and improve the safety of cold-cut meats. The combination of caffeic acid and ferulic acid in a 1:1 ratio demonstrated a synergistic effect. It was also reported that caffeic acid showed antimicrobial potential and/or synergistic effects with antibiotics against *S. aureus*, *S. epidermidis*, *K. pneumoniae*, *S. marcescens*, *P. mirabilis*, *E. coli*, *P. aeruginosa*, *B. cereus*, *M. luteus*, *L. monocytogenes*, and *C. albicans* strains [110]. When combined, phenolic acids could reflect considerably higher antimicrobial efficacy against foodborne pathogens compared to individual phenolic acids [111].

The beneficial effects of oats consumption on the conditions and diseases of the digestive tract have also been manifested by other mechanisms. Seether et al. [112] showed that oat OH alcohol extract has the potential to inhibit gastric acid secretion in pigs. Moreover, some physiological benefits of oatmeal were reported, such as reducing hypercholesterolemia, hyperglycemia, hyperinsulinemia, and hypertension. Those activities are often attributed to *β*-glucans and phenolics [9].

Based on our findings and all of the above facts, we can conclude that there is the potential for developing oat-based products that are beneficial to the digestive tract.

## 4. Conclusions

This study investigated the nutritional and bioactive properties of oat flour (OF) and ground oat husks (OHs) from three differently colored oat genotypes. Both OF and OHs were rich in essential nutrients, particularly protein, lipids, and dietary fiber. The phenolic profile analysis revealed a significant presence of phenolic acids, primarily ferulic acid, in both OF and OHs. OHs had 13 to 25 times higher contents of total insoluble-bound phenolic compounds. These bioactive compounds may contribute to the strong antioxidant and antimicrobial activities observed in the extracts. Notably, OH extracts exhibited superior antimicrobial properties against several gastrointestinal pathogens, *L. monocytogenes*, *E. coli*, and *S. haemolyticus*. These findings highlight the potential of OF and OHs as valuable ingredients in functional foods and nutraceuticals. The high nutritional content and the presence of bioactive compounds suggest that both the main constituent—oat grain as well as the byproduct—husk can offer numerous health benefits, such as improved gut health, reduced oxidative stress, and enhanced immunity. However, further research is needed to fully elucidate the mechanisms underlying these effects and to optimize the extraction procedures and formulation of oat-based products.

## Figures and Tables

**Figure 1 foods-13-03853-f001:**
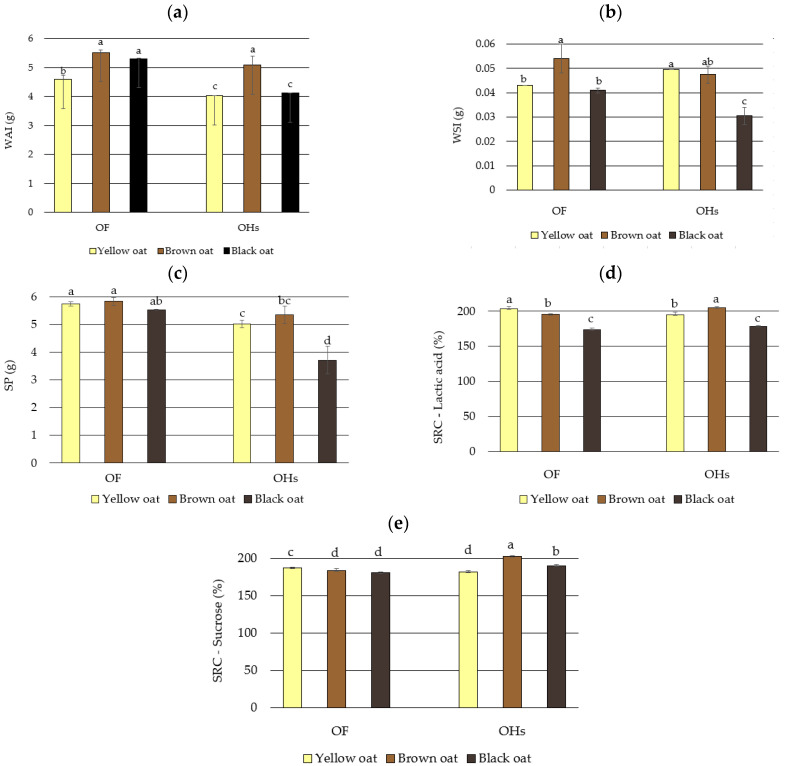
Gelling properties: (**a**) water absorption index; (**b**) water solubility index; (**c**) swelling power, and solvent retention capacity (**d**) lactic acid SRC; (**e**) sucrose SRC of oat flour (OF) and oat husks (OHs). Values are means of three determinations ± standard deviation. Means followed by the same letter within the same column are not significantly different (*p* < 0.05).

**Figure 2 foods-13-03853-f002:**
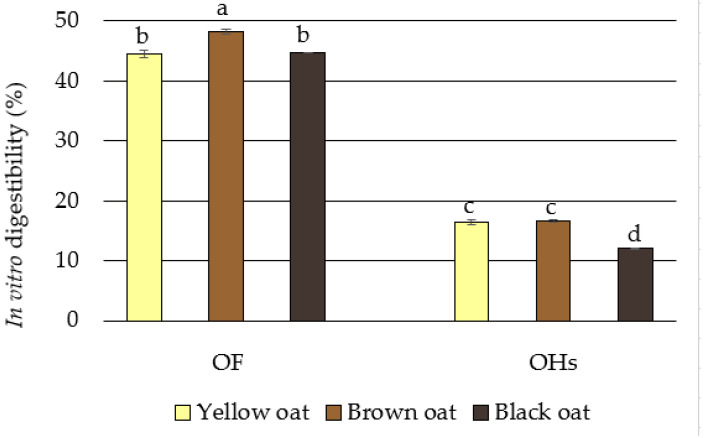
In vitro digestibility of oat flour (OF) and oat husks (OHs). Values are means of three determinations ± standard deviation. Means followed by the same letter within the same column are not significantly different (*p* < 0.05).

**Table 1 foods-13-03853-t001:** Basic chemical composition of OF and OHs.

Sample	Dry Matter	Total Starch	Total Protein	Alcohol-Soluble Proteins		Fat	Ash
	(%)	(% d.m.)	(% d.m.)	(% d.m.)	(% t.p.)	(% d.m.)	(% d.m.)
OF							
Yellow	90.60 ^e^	55.71 ± 0.01 ^b^	17.38 ± 0.10 ^a^	2.04 ± 0.05 ^b^	11.73	5.45 ± 0.16 ^c^	2.44 ± 0.01 ^d^
Brown	90.92 ^d^	57.38 ± 0.54 ^a^	16.67 ± 0.09 ^b^	2.31 ± 0.11 ^a^	13.86	7.20 ± 0.14 ^a^	2.30 ± 0.02 ^e^
Black	90.99 ^d^	57.58 ± 1.15 ^a^	13.43 ± 0.01 ^c^	1.60 ± 0.01 ^c^	11.91	6.15 ± 0.21 ^b^	2.19 ± 0.04 ^f^
OHs							
Yellow	96.06 ^a^	n.d.	4.79 ± 0.06 ^d^	0.64 ± 0.09 ^d^	13.36	1.23 ± 0.18 ^d^	5.85 ± 0.02 ^a^
Brown	94.69 ^b^	n.d.	4.54 ± 0.02 ^e^	0.64 ± 0.10 ^d^	14.10	1.02 ± 0.05 ^d^	4.64 ± 0.00 ^b^
Black	94.51 ^c^	n.d.	3.09 ± 0.02 ^f^	0.56 ± 0.01 ^d^	18.12	1.32 ± 0.06 ^d^	5.00 ± 0.00 ^b^

OF—oat flour; OHs—oat husks; t.p.—total protein. Values are means of three determinations ± standard deviation. Means followed by the same letter within the same column are not significantly different (*p* < 0.05).

**Table 2 foods-13-03853-t002:** Content of dietary fibers in OF and OHs (% d.m.).

Sample	NDF	ADF	ADL	Hemicellulose	Cellulose	*β*-Glucan	Resistant Starch
OF							
Yellow	15.83 ± 0.31 ^f^	3.11 ± 0.08 ^d^	0.89 ± 0.20 ^d^	12.72 ± 0.40 ^e^	2.22 ± 0.28 ^d^	5.33 ± 0.01 ^a^	2.95 ± 0.39 ^b^
Brown	32.36 ± 0.25 ^d^	3.65 ± 0.12 ^d^	1.29 ± 0.47 ^cd^	28.71 ± 0.35 ^c^	2.36 ± 0.38 ^d^	4.07 ± 0.16 ^c^	2.43 ± 1.03 ^a^
Black	17.22 ± 0.04 ^e^	3.34 ± 0.26 ^d^	0.89 ± 0.17 ^d^	13.88 ± 0.23 ^d^	2.45 ± 0.09 ^d^	4.66 ± 0.00 ^b^	2.91 ± 0.50 ^b^
OHs							
Yellow	80.27 ± 0.08 ^b^	38.96 ± 0.54 ^a^	6.00 ± 0.11 ^a^	41.31 ± 0.62 ^b^	32.96 ± 0.43 ^c^	0.06 ± 0.00 ^d^	n.d.
Brown	77.91 ± 0.24 ^c^	36.04 ± 0.69 ^c^	1.86 ± 0.28 ^b,c^	41.87 ± 0.45 ^b^	34.18 ± 0.42 ^b^	0.06 ± 0.00 ^d^	n.d.
Black	83.82 ± 0.22 ^a^	37.71 ± 0.08 ^b^	2.39 ± 0.26 ^b^	46.11 ± 0.13 ^a^	35.32 ± 0.18 ^a^	0.03 ± 0.01 ^d^	n.d.

OF—oat flour; OHs—oat husks; NDF—neutral detergent fiber, ADF—acid detergent fiber, ADL—acid detergent lignin. Values are means of three determinations ± standard deviation. Means followed by the same letter within the same column are not significantly different (*p* < 0.05).

**Table 3 foods-13-03853-t003:** Content of phenolic compounds and antioxidant capacity of OF and OHs.

Sample	TPC (μg GAE/g)	*p*-Coumaric Acid (μg/g)	Ferulic Acid (μg/g)	Isoferulic Acid (μg/g)	Vanillic Acid (μg/g)	Syringic Acid (μg/g)	TAC (mmol Trolox Eq/kg)
OF							
Yellow	875.92 ± 11.73 ^d^	90.96 ± 8.98 ^c^	395.88 ± 13.72 ^d^	n.d.	23.05 ± 0.48 ^d^	18.14 ± 0.83 ^c^	23.85 ± 1.46 ^cd^
Brown	841.89 ± 54.16 ^d^	82.1 ± 8.67 ^c^	448.72 ± 20.65 ^d^	8.35 ± 1.07 ^a^	22.74 ± 0.95 ^d^	19.76 ± 1.34 ^c^	24.28 ± 1.36 ^c^
Black	982.08 ± 43.69 ^d^	84.40 ± 5.10 ^c^	589.14 ± 13.01 ^d^	10.96 ± 0.67 ^a^	16.66 ± 0.72 ^d^	19.47 ± 0.79 ^c^	22.05 ± 1.01 ^d^
OHs							
Yellow	12,086.76 ± 259.41 ^b^	6732.36 ± 325.83 ^a^	4987.02 ± 110.36 ^c^	29.11 ± 0.94 ^a^	101.44 ± 5.53 ^c^	65.54 ± 0.73 ^b^	37.32 ± 2.03 ^b^
Brown	21,971.52 ± 890.64 ^a^	2749.63 ± 30.12 ^b^	13,794.82 ± 122.57 ^a^	49.85 ± 57.29 ^a^	442.62 ± 0.97 ^b^	92.79 ± 5.36 ^a^	48.19 ± 0.11 ^a^
Black	24,352.48 ± 528.41 ^c^	2726.76 ± 60.14 ^b^	13,271.2 ± 59.1 ^b^	70.59 ± 6.55 ^a^	619.38 ± 8.26 ^a^	63.25 ± 6.85 ^b^	46.64 ± 1.01 ^a^

OF—oat flour; OHs—oat husks; Means followed by the same letter within the same column are not significantly different (*p* < 0.05). TPC—total phenolic compounds; TAC—total antioxidant capacity. Values are means of three determinations ± standard deviation. Means followed by the same letter within the same column are not significantly different (*p* < 0.05).

**Table 4 foods-13-03853-t004:** Minimal bactericidal concentration (MBC) of hydrolyzed and traditionally prepared oat extracts expressed in microliters of the preparation.

Strain	Yellow		Brown		Black	
	OF	OHs	OF	OHs	OF	OHs
Hydrolyzed samples						
*Listeria monocytogenes*	<140	<140	200	160	180	140
*Staphylococcus haemolyticus*	140	140	>200	>200	>200	>200
*Salmonella typhimurium*	>200	180	>200	200	>200	>200
*Enterococcus faecalis*	140	140	>200	180	180	>200
*Escherichia coli*	140	140	200	140	140	140
*Shigella flexneri*	160	150	>200	180	>200	>200
Traditionally prepared samples						
*Listeria monocytogenes*	>200	>200	>200	160	220	150
*Staphylococcus haemolyticus*	180	150	>200	150	>200	150
*Salmonella typhimurium*	>200	>200	>200	200	>200	200
*Enterococcus faecalis*	>200	200	>200	180	>200	180
*Escherichia coli*	180	180	220	160	180	160
*Shigella flexneri*	220	180	>200	180	>200	160

OF—oat flour; OHs—oat husks.

## Data Availability

The original data presented in the study are openly available in RIK at https://rik.mrizp.rs/?locale-attribute=sr_RS accessed on 22 November 2024.

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
