# Peer review of "Characterization and Potential Food Applications of Oat Flour and Husks from Differently Colored Genotypes as Novel Nutritional Sources of Bioactive Compounds"

_foods, 2024, doi:10.3390/foods13233853_

Round 1

Reviewer 1 Report

Comments and Suggestions for Authors

Dear Editors and Authors

The presented article is very comprehensive and includes the nutritional properties, phenolic profile and baking functionality of oat flour (OF )and ground oat husks (OH) from white, brown and black oat genotypes, as well as the antioxidant and antimicrobial activities of their extracts. The authors showed that OF and OH were rich in essential nutrients, protein, lipids and dietary fiber. The antioxidant and antimicrobial activity observed in extracts from OH and OF is related to the presence of phenolic compounds. Oat by-products may offer numerous health benefits: improved intestinal health, reduced oxidative stress and increased immunity. They can therefore be valuable ingredients in functional foods and nutraceuticals. 

minor revisions

Line 127-128 Streptomyces griseus should be written in italics

Line 148 the oil or the fat? Table 1, Line 354.356,357, 362,365

Line 238-240 “Standards of gallic acid, 3,4-dihydroxybenzoic acid, chlorogenic acid, vanillic acid, caffeic  acid, siringic acid, p-coumaric acid, sinapic acid, ferulic acid, and isoferulic acid were used  (10, 20, 40, 50 and 100 μg/g)” unit μg/g is correct? Standard were purchased from??(no data) HPLC method validation parameters such as limits of detection or quantification are not provided, if previously published indicate source

Line 239 siringic acid? according to the reviewer syringic acid

Line 306-307 Listeria monocytogenes, Staphylococcus haemolyticus, Salmonella typhimurium, Enterococcus faecalis, Escherichia coli, Shigella flexneri should be written in italics

Line 328 the Authors write” ..the data were reported as a mean ± standard deviation of at least three independent repetitions per sample” but earlier in line 219-220 the authors write ”.. all extractions were performed in duplicate per each sample and kept at -70 °C before analyses”-please explain?

Table 2 should be % or %/d.m.??  2.3.3. Analysis of dietary fibers Line 180 -190 Authors write“….all the results are given as the percent per d.m.” Abbreviations NDF, ADF and ADL should be explained below the table

Tables 1,2 and 3 the abbreviations OF and OH should be explained below the table and standardized, justified or to the left

Table 4 the abbreviations OF and OH should be explained below the table

Line 402 According to the reviewer antioxidant effect of dietary fibre is based on the phenolic compounds bound to polysaccharide complexes

Reviewer 2 Report

Comments and Suggestions for Authors

Personally, I find the text interesting. This study promotes the utilization of a common plant material. However, the manuscript presents a series of drawbacks that need to be corrected. In the following lines I will explain the main mistakes found.

Line 15. “β” must be in italics.

Line 43-45. Rewrite.

Line 46. “Avena nuda L” This species is not mentioned again in the entire text.

Line 47. Give some examples.

Line 48-50. Do you know anything about its traditional medical use in other countries?

Line 51-54. It seems that in your study you are going to use by-products, but in the end it is not carried out. Assess how to put this information in the most appropriate way so as not to lose sight of the objective of the study.

The introduction must be restructured. In its current state, it has a considerable lack of organization. For example, in a paragraph we begin to talk about the composition of oats and end by talking about their biological activities. In the next paragraph we talk about composition again, and then talk about bioactivities again. The work is interesting, but it has a notable lack of revision.

Chemicals section. Only methanol is of HPLC grade? Where is the information about standars?

Section 2.3.1. It is noted again that the authors did not take the necessary time to review their manuscript. There are basic writing errors. For example, there are many punctuation errors. These errors are not admissible.

Section 2.3.1. what about carbohydrates? And moisture content?

Line 150. “650 °C” instead of “650°C”. Correct this error throughout the entire manuscript. Sometimes it appears correctly, but many times it does not. This again demonstrates negligence on the part of the authors.

Something similar happens with volume units. It is mL, with the L in capital letters. But the unit is in a different format with only three words difference between them! How can you not detect those errors! It is clear that the authors did not make the necessary effort. Errors that are so easily resolved but that the authors have not previously reviewed make it meaningless to assess the quality of the work. It cannot be that in a review you have to say in which lines there are missing points, in which others there is a lack of formatting, etc.

Line 194. Again starch?

Line 248. •+ as superscript.

Line 256-258. Rewrite.

Line 290. Latin expressions must be in italics.

Line 307. Scientific names must be in italics.

Not even the tables or the figures are well formatted.

Line 343. These are not values from the table.

There is no data on the chromatograms or how the identification of the compounds was carried out.

There are sections of the discussion that have to be discussed more widely. Furthermore, the level of writing is poor. Sometimes it is necessary to read several times to understand what you are trying to say.

The authors use a lot of self-citations.

Comments on the Quality of English Language

The English level must be improved.

Round 2

Reviewer 2 Report

Comments and Suggestions for Authors

Authors have carried out all the suggested changes.